# Identification of Male-Specific Molecular Markers by Recombination of *RhoGEF10* Gene in Spotted Knifejaw (*Oplegnathus punctatus*)

**DOI:** 10.3390/genes13071262

**Published:** 2022-07-15

**Authors:** Yanduo Wu, Yongshuang Xiao, Zhizhong Xiao, Yuting Ma, Haixia Zhao, Jun Li

**Affiliations:** 1CAS and Shandong Province Key Laboratory of Experimental Marine Biology, Center for Ocean Mega-Science, Institute of Oceanology, Chinese Academy of Sciences, Qingdao 266071, China; ydwu_1902448082@163.com (Y.W.); yongshuangxiao@qdio.ac.cn (Y.X.); xzz@qdio.ac.cn (Z.X.); mayuting1998@163.com (Y.M.); zhaohaixia9517@163.com (H.Z.); 2Southern Marine Science and Engineering Guangdong Laboratory (Guangzhou), Guangzhou 511458, China; 3Laboratory for Marine Biology and Biotechnology, Qingdao National Laboratory for Marine Science and Technology, Qingdao 266071, China; 4School of Oceanography, University of Chinese Academy of Sciences, Beijing 100049, China; 5School of Marine Science and Engineering, Qingdao Agricultural University, Qingdao 266109, China

**Keywords:** *Oplegnathus punctatus*, *RhoGEF10* gene, insertion/deletion polymorphism, sex molecular marker

## Abstract

The spotted knifejaw (*Oplegnathus punctatus*) is a marine economic fish with high ecological value, food value, and fishing value, and its growth has obvious sex dimorphism. The rapid identification of its sex is beneficial to the development of sex determination and breeding. In this study, the method of comparative genomics and PCR amplification was used to further establish a rapid detection method for the recombinant *RhoGEF10* gene in *O. punctatus*, which can quickly, accurately, and efficiently identify the sex of the *O. punctatus* to be tested. The homologous comparison results of male and female individuals showed that the DNA fragment length of the *RhoGEF10* gene on the X1 chromosome was 326 bp, and the DNA fragment length on the Y chromosome was 879 bp. Therefore, it can be concluded that there is an insert fragment of 553 bp on the Y chromosome. PCR amplification results showed that the two DNA fragments of 879 bp and 326 bp were amplified in the Y chromosome and X1 chromosome of the male *O. punctatus* (X1X2Y), respectively, and the 879 bp fragment was a unique marker fragment of the recombinant *RhoGEF10* gene; The female *O. punctatus* (X1X1X2X2) only a single DNA fragment of 326 bp was amplified. At the same time, the inserted fragment of the male individual resulted in partial inactivation of the RhoGEF10 protein, which in turn resulted in a slowing of peripheral nerve conduction velocity and thinning of the myelin sheath in male *O. punctatus*. The method shortens the time for accurate identification of the *O. punctatus* *RhoGEF10* gene recombination and improves the detection efficiency. It is of great significance and application value in the research of nerve conduction and myelin development, male and female sex identification, the preparation of high male seedlings, and family selection based on the *RhoGEF10* gene in the *O. punctatus.*

## 1. Introduction

The spotted knifejaw, *Oplegnathus punctatus*, belongs to Perciformes, Oplegnathidae, and Oplegnathus, and is mainly distributed in the temperate zone waters of the western Pacific Ocean [1]. At the same time, due to the shell-feeding characteristics of *O. punctatus*, it is possible to clean the netting of aquaculture nets, which saves a lot of human resources and netting cleaning costs, and the economic and ecological benefits are extremely significant [2]. In China, the Institute of Oceanography, Chinese Academy of Sciences, together with Shandong Laizhou Mingbo Aquatic Co., Ltd., made the first breakthrough in the reproductive control and full artificial breeding technology of *O. punctatus* parents in 2014, making *O. punctatus* an important new object for seawater netting and factory farming, with the current market price reaching 200 RMB/catty [3]. Since *O. punctatus* is a native natural species in China, the development and sustainable use of its resources are more long-lasting, and the breeding prospects are extremely broad. The karyotype of female *O. punctatus* is 2*n* = 48 with genetic sex type (X1X1X2X2), and the karyotype of the male is 2*n* = 47 with one large metacentric Y chromosome in male individuals of *O. punctatus* chromosome and genetic sex type (X1X2Y) [4,5,6]. The spotted knifejaw has significant male and female growth dimorphism in the breeding process, male individuals grow faster than female fish, the preparation of *O. punctatus* Kaohsiung fry and the all-male fry will promote the quality and efficiency of *O. punctatus* breeding industry, and the development of rapid genetic sex identification technology for grouper fry will help the reserve of excellent male germplasm of *O. punctatus* [7].

Rho multidomain guanine nucleotide exchange factors (RhoGEF) are key factors in the regulation of Rho GTPase activity and play a crucial role in the regulation of Rho GTPase signaling, growth and development of tissues and organs, and immune response [8,9]; they are also closely related to tumors, developmental and neurological diseases, and pathogenic microbial infections, etc. [10,11]. RhoGEF consists of two major categories, namely, Diffuse B lymphoma (Dbl) family multidomain guanine nucleotide exchange factors, and non-Dbl multidomain guanine nucleotide exchange factors, of which the more studied are the Dbl family GEFs [12,13]. All Dbl family members share a common feature of containing DH and PH structural domains [14]. RhoGEF activates Rho GTPase conformational changes by catalyzing the exchange of bound GDP and GTP, thus prompting its interaction with downstream effector proteins [15]. Recent evidence suggests that Rho GTPases are involved in neuronal morphogenesis, including cell migration, axonal growth and guidance, dendrite refinement and plasticity, and synapse formation, and play a central role in defining the spatiotemporal activation of the corresponding GTPases within neuronal cells [16,17].

Insertion/deletion polymorphism is a molecular marker, which is the second most frequent alteration at the DNA or protein sequence level after residue substitution, mainly manifested by the insertion or deletion of fragmented DNA of different sizes in the genome [18]. As an important member of the *RhoGEF* family, *RhoGEF10* is involved in the normal development of peripheral nerves in vertebrates, influencing peripheral nerve conduction velocity and myelin sheath thickness [15]. However, there are no reports on the recombination detection method of *RhoGEF10* gene insertion on male chromosomes of *O. punctatus* and its application to the genetic sex identification of females and males.

## 2. Materials and Methods

### 2.1. Genome-Wide Identification and Characterization of RhoGEF10

To identify the *RhoGEF10* gene of *O. punctatus* (CNGB accession code: CNP0001488) [5], a local Blast database was established and combined with Hmm search results. The colinearity of the *RhoGEF10* gene on the chromosomes and the electrophoresis pattern were determined by the TBtools software (https://github.com/CJ-Chen/TBtools/releases, accessed on 21 April 2022). The selection condition of collinearity is that the base length is greater than 1 kb, and the base sequence similarity is 99%. Amino acid sequence alignment using Multiple Alignment using Fast Fourier Transform (MAFFT) (https://www.ebi.ac.uk/Tools/msa/mafft/, accessed on 6 May 2022). The *RhoGEF10* gene structure pattern of male and female *O. punctatus* was mapped by IBS software (http://ibs.biocuckoo.org/online.php, accessed on 31 May 2022).

### 2.2. Source of Experimental Samples

Take the artificial breeding spotted knifejaw, including the XX females and XY males. Experimental *O. punctatus* was obtained from Haihe Turbot Breeding Co., LTD. in Wenden, Weihai (Shandong, China). A total of 22 sexually mature individuals (11 males and 11 females) of *O. punctatus* were collected for genomic DNA extraction.

### 2.3. Genomic DNA Extraction

The genomic DNA of the XX female and XY male mentioned above was extracted using the TIANGEN (Hengfei, China) marine animal DNA extraction kit to extract the muscle DNA of the *O. punctatus* according to the manufacturer’s instructions. Identify the integrity of the genomic DNA by 1.0% agarose gel electrophoresis, measure the OD value of the DNA supernatant with a UV spectrophotometer (NanoDrop2000, Thermo Scientific, Chicago, USA) (OD260 nm OD280 nm ≥ 1.8, OD260 nm OD230 nm ≥ 1.8 stand for good quality) [19], and adjust the DNA concentration to 50 ng/µL, frozen at −20 °C for later use. The genomic DNA was stored at 4 °C.

### 2.4. Gender Molecular Markers in Spotted Knifejaw

Through comparative genomic bioinformatics analysis, it was found that the *RhoGEF10* gene demonstrated differences [5]. Then, the genomic DNA of each *O. punctatus* was used for PCR amplification to detect the known biological sex by specific primers designed from the *RhoGEF10* gene sequence, the upstream and downstream primer sequences are: ChRho_F1: 5′-CAACTACCACTCTGAATGCTGC-3′; ChRho_R2: 5′-GTTCAACCACCACACTTTAGGC-3′. The specific primers were synthesized by Tsingke Biotechnology Co., Ltd. (Qingdao, China).

The specific marker DNA fragment inserted on the *RhoGEF10* gene of the Y chromosome was amplified by PCR technology. The recombinant *RhoGEF10* gene was detected by PCR method using the DNA fragment primers ChRho_F1 and ChRho_R2 specific for the recombinant *RhoGEF10* gene. The PCR reaction system is: 20 µL, including 10 × Buffer 4.0 µL; dNTP (2.5 mmol/L) 3.0 µL; rTaq enzyme (TransGen Biotech, Beijing, China) (5 U/µL) 0.2 µL; ChRho_F1 1.0 µL; ChRho_R2 1.0 µL; DNA template 1.0 µL, 9.8 µL dd H_2_O; the above liquids were mixed and centrifuged. Touchdown PCR amplification program was: 95 °C 3 min, 61 °C 30 s, 72 °C 1 min 30 s; 95 °C 3 min, 60 °C 30 s, 72 °C 1 min 30 s; 95 °C 3 min, 59 °C 30 s, 72 °C 1 min 30 s; 95 °C 30 s, 58 °C 30 s, 72 °C 1 min 30 s, 28 cycles; 72 °C 10 min, at last, hold the temperature at 15 °C. The PCR product was detected by 1.5% agarose gel electrophoresis to quickly determine whether the *RhoGEF10* gene of the tested spotted knifejaw has undergone insertion and recombination; at the same time, this marker is located on the Y chromosome of male fish, with male-associated genetic characteristics, and is applied to accurate identification of genetic sex of spotted knifejaw. Then, the recombinant and non-recombinant *RhoGEF10* gene differential fragment products were recovered by gel tapping and transformed into competent cells (TransGen Biotech, Beijing, China) through the PMD18-T vector (Takala, Tokyo, Japan). Positive clones were picked and sent to Qingdao Paisennuo Gene Technology Co., Ltd. (Qingdao, China) for sequencing.

## 3. Results

### 3.1. Sequence Analysis of RhoGEF10 Gene

The study results showed that the nucleotide sequence length of the female *RhoGEF10 gene* was 326 bp, while that of the male *RhoGEF10* gene was 879 bp. An insertion of 553 bp in length was found on the Y chromosome (Figure 1).

### 3.2. Homologous Comparison of RhoGEF10 Gene in Males and Females

As shown in Figure 2, the *RhoGEF10* gene is located on the X1 chromosome in females and on the Y chromosome in males. Figure 2 also shows a recombinant insertion of a large segment of the DNA sequence in the homologous *RhoGEF10* gene on the Y chromosome of the male *O. punctatus*.

Through homologous alignment, it was found that the homologous *RhoGEF10* gene on the Y chromosome of the male *O. punctatus* had a 3681 bp DNA base sequence recombination insertion. We selected the 60,000 bp~65,000 bp region of the *RhoGEF10* gene on the X1 chromosome of female fish and the 75,000 bp~80,000 bp region of the Y chromosome of male fish, which is homologous to it and has insertional recombination, as the target regions for our study. The *RhoGEF10* gene on the X1 chromosome of the female fish is homologous to the Y chromosome of the male fish in the 60,000 bp~65,000 bp region, and there is insertion recombination in the 75,000 bp~80,000 bp region in the male fish. At the same time, the DNA fragment on the X1 chromosome is 326 bp in length, named ChX1Rho; the DNA fragment on the Y chromosome is 879 bp in length, including the homologous sequence (326 bp) and the recombinant insertion sequence (553 bp) of the *RhoGEF10* gene on the Y chromosome, named ChY Rho (Figure 3).

We can conclude that a DNA sequence insertion of size 553 bp was found between the 55th and 56th positions on ChY Rho by aligning with the homologous sequence of ChX1Rho. Figure 4 is a pattern diagram of the insertion of 553 bp DNA fragments between the nucleotide sequences of ChX1Rho of the X chromosome and ChYRho of the Y chromosome (Figure 4).

It can be seen from Figure 5 that the amino acid length of RhoGEF10 protein in female *O. punctatus* is 1296 bp, while the amino acid length of RhoGEF10 protein in male individuals is only 93 bp. The insertion of nucleotide base fragments leads to the shortening of the amino acid fragment of male RhoGEF10.

### 3.3. Genetic Sex Identification of New Markers in O. punctatus

The PCR results showed that two DNA fragments of 879 bp and 326 bp were amplified in the Y chromosome and X1 chromosome of the male *O. punctatus* (X1X2Y), respectively, and the 879 bp fragment was a unique marker fragment of the recombinant *RhoGEF10* gene; only a single DNA fragment of 326 bp was amplified in the female individual (X1X1X2X2), as shown in Figure 6b. Figure 6a is the electrophoresis pattern. Since the ChYRho marker is located on the Y chromosome of the male fish, it has the genetic characteristics of a co-male, so the ChYRho fragment is also a DNA marker unique to the Y chromosome of the *O. punctatus*, and ChX1Rho is a genetic marker of the sex of the *O. punctatus*.

The individual showing two bands (326 bp and 879 bp) is a recombinant individual of the *RhoGEF10* gene, which is also a genetic male and is identified as male by physiological sex histology, while the individual showing a single band (326 bp) is a non-recombinant individual of *RhoGEF10* gene, which is a genetic female and is identified as female by physiological sex histology.

## 4. Discussion

The growth of fish is one of the most important economic traits of interest in aquaculture today, and the phenomenon of sexual dimorphism in its growth exists in much economic fish [20]. The growth rate and individual size vary greatly between males and females. For example, in blotched snakehead (*Channa maculata*) [21] and Nile tilapia (*Oreochromis niloticus* L.) [22], the growth rate of male blotched snakeheads is significantly higher than that of females, while in European eel (*Anguilla anguilla*) [23]) and half-smooth tongue sole (*Cynoglossus semilaevis*) [24], the growth rate of females is higher than that of males so that the production of all-male or all-female fry can significantly improve the culture efficiency. In *O. punctatus*, the growth rate of males is generally higher than that of females, which is similar to that of blotched snakehead and Nile tilapia. Meanwhile, the sex determination mechanism of *O. punctatus* belongs to the multiple sex chromosome system X1X1X2X2/X1X2Y, so some scholars speculate that this mechanism is related to its growth dimorphism [25]. Because of its growth dimorphism, the accurate identification of the genetic sex of the grouper is one of the important bases for its sex control technology, and the specific sex molecular markers are an important tool for genetic sex identification. In recent years, the study of sex-linked molecular markers in fish has gradually increased, such as in bighead carp (*Hypophthalmichehys nobilis*) and silver carp (*Hypophthalmichthys molitrix*) [26], and northern snakehead (*Channa argus*) [27], where researchers have used simplified genome sequencing to screen for sex-linked molecular markers.

Based on a comparative analysis of the re-sequenced genomes, a 15 bp deletion in the third intron of the *Dmrt1* gene was finally screened between yellow croaker males and females of large yellow croaker (*Larimichthys crocea*) [28]. In rock bream (*Oplegnathus fasciatus*), Xu identified a male barred knifejaw specific marker with an 8 bp deletion and several single nucleotide polymorphism (SNP) positions by AFLP method, based on which specific primers applicable for genetic sex identification were developed [29]. In *O. punctatus*, which also has heteromorphic chromosomes as rock bream, a comparative analysis of the female and male genomes was performed by Ming Li et al., and an 18 bp insertion site was identified in the male genome, after validation by the sequencing depth and PCR. Then, an effective PCR-based method was developed to identify the sex of *O. punctatus*.

In this study, however, insertional recombination of a 553 bp-sized DNA fragment in the *RhoGEF10* gene on the Y chromosome of male *O. punctatus* was found by comparative genomics and DNA sequence alignment analysis, and a DNA marker specific to the insertional recombination of the *RhoGEF10* gene on the Y chromosome of grouper was designed to establish a PCR for rapid identification of genetic sex in the *O. punctatus RhoGEF10* gene recombination. A PCR method was developed for the rapid genetic identification of *RhoGEF10* recombination in *O. punctatus*. The method amplified two bands of 879 bp and 326 bp in the recombinant *RhoGEF10* individuals, of which the 879 bp band was the specific target band, while only a single band (326 bp) was amplified in the non-recombinant *RhoGEF10* individuals, and the above target bands could be distinguished quickly and accurately by agarose gel electrophoresis, and the recombination of *RhoGEF10* gene could be quickly identified. Meanwhile, this specific target band (879 bp) is located on the Y chromosome of male fish, which is a companion male genetic characteristic, so this fragment is also a unique DNA marker for the Y chromosome of male grouper and can be used for rapid identification of female and male genetic traits in *O. punctatus*.

Insertional mutations, that is, mutations in the DNA strand caused by the insertion of additional nucleotides or DNA fragments [30]. Insertion mutations occur from time to time, and in humans, exon 20 insertion mutations lead to an increased prevalence of non-small-cell lung cancer (NSCLC) [31]; in the medaka fish (*Oryzias latipes*), intra-exon insertion of an additional 1.9 kb fragment resulted in oculocutaneous albinism [32]; in human neonatal fetuses, we found that FGFR3 insertions resulted in shortened extremities, curved femurs, and narrowed thorax. This mutation also leads to overexpression of Fgfr3 in zebrafish, resulting in downstream signaling and overactivation of the dorsal embryo [33]. In this study, the *RhoGEF10* gene on the male chromosome of the *O. punctatus* was inserted with a nucleotide sequence of 553 bp in length relative to the female individual, which resulted in the amino acid sequence length of the male individual being only 93 bp. It is speculated that in male individuals, the RhoGEF10 protein is partially inactivated, and this gene is associated with peripheral nerve conduction velocity and myelin thickness [34,35], which infers peripheral nerve conduction velocity and myelin thinning in male fish.

## 5. Conclusions

We designed a specific DNA marker for the insertional recombination of the *RhoGEF10* gene on the Y chromosome of the male *O. punctatus*, using the insertional recombination of the 553 bp-sized fragments of DNA on the Y chromosome in the *O. punctatus*, and successfully identified the marker for the genetic sex identification of the *O. punctatus*. This method is an efficient, rapid, and accurate method to identify whether recombination occurs in the *RhoGEF10* gene of *O. punctatus*, which is of great significance and application to reveal the differences in peripheral nerve conduction patterns and myelin development between female and male *O. punctatus*. This study was conducted to achieve rapid identification of genetic sex between females and males and to enhance the genetic breeding process and large-scale production of high-quality fry of *O. punctatus*.

## Figures and Tables

**Figure 1 genes-13-01262-f001:**
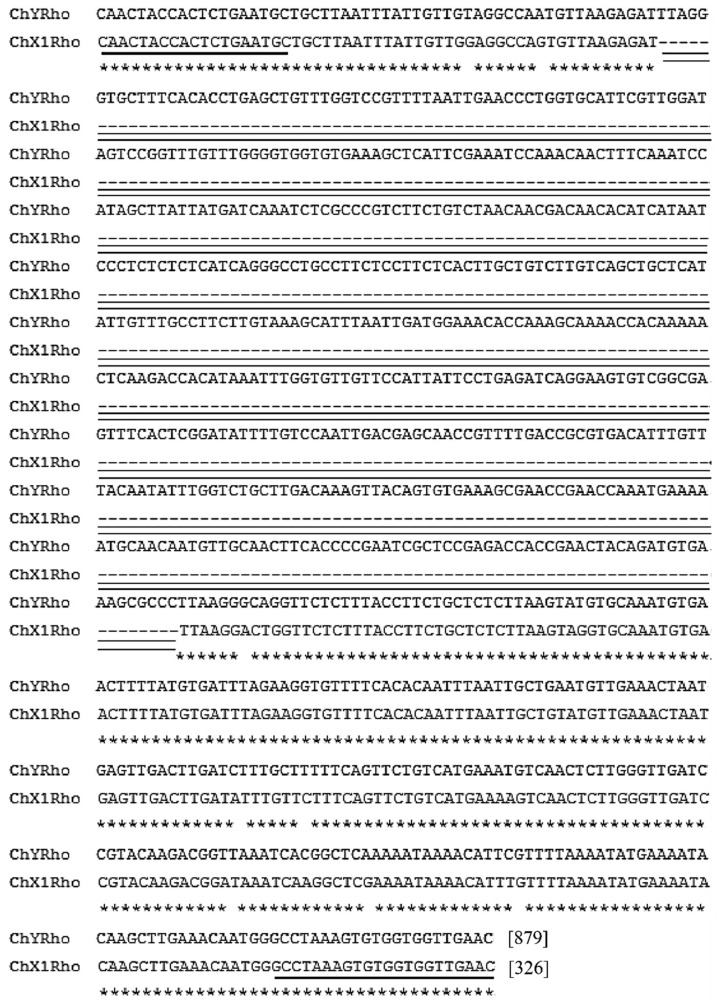
Nucleotide sequence alignment of the X chromosome *RhoGEF10* gene with the Y chromosome *RhoGEF10* gene (Appendix A). Note: The nucleotide sequence comparison between chromosome X (ChX1Rho) and chromosome Y (ChYRho), the primer positions, are underlined in a bold single horizontal line; *: represents the sequence identity of ChX1Rho and ChYRho, and the blank area indicates the base deletion or inconsistent sequence of both; ----: represents the insertion of the missing sequence; the unbolted double dash represents the region where the missing sequence is located.

**Figure 2 genes-13-01262-f002:**
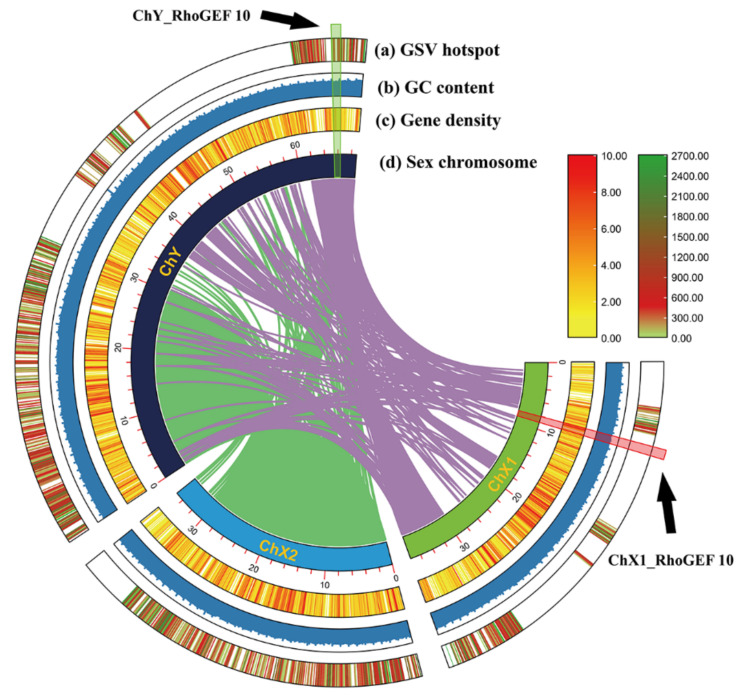
Chromosome colinearity map constructed from the sequence similarity of chromosomal DNA in male and female *O. punctatus*. Note: (**a**) GSV hotspot: Genomic structure variation hotspot, (**b**) GC content, (**c**) Gene density, (**d**) Sex chromosome. The black arrows in the figure point to the location of the *RhoGEF10* gene on the Y chromosome and the location of the *RhoGEF10* gene on the X1 chromosome.

**Figure 3 genes-13-01262-f003:**
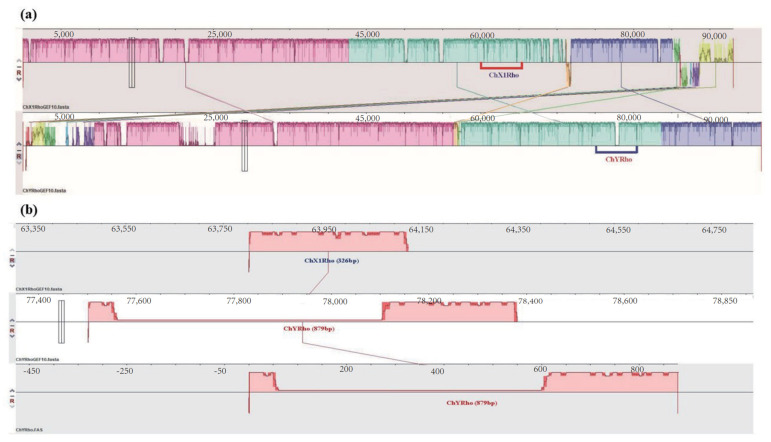
Information on the specific location of ChX1Rho and ChYRho on the X and Y chromosomes, respectively. Note: (**a**): The same color modules in the figure represent highly homologous regions of the *RhoGEF10* gene on the X and Y chromosomes, where the underlined red and blue boxed regions are where the target sequences of the present invention, ChX1Rho and ChYRho, are located. (**b**): 326 bp represents the length of the ChX1Rho fragment and 879 bp represents the length of ChYRho; the pink region in the lower middle of ChYRho represents the inserted 553 bp nucleotide sequence, which has no homologous matching sequence with ChX1Rho fragment.

**Figure 4 genes-13-01262-f004:**
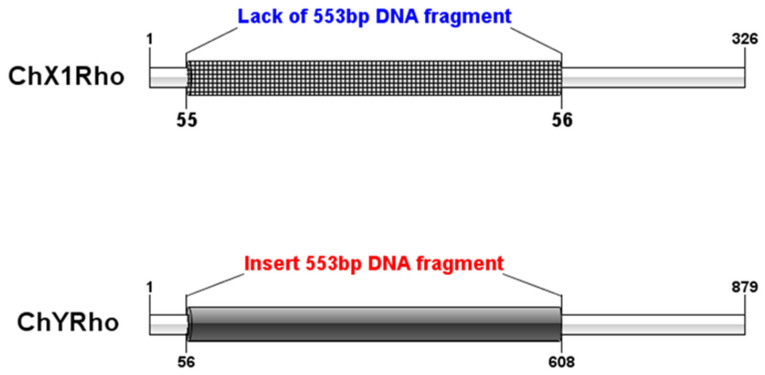
The pattern diagram of the homologous insertion of the nucleotide sequences of ChX1Rho of the X chromosome and ChYRho of the Y chromosome. Note: the same color regions represent homologous regions and different colors represent insertion site information.

**Figure 5 genes-13-01262-f005:**
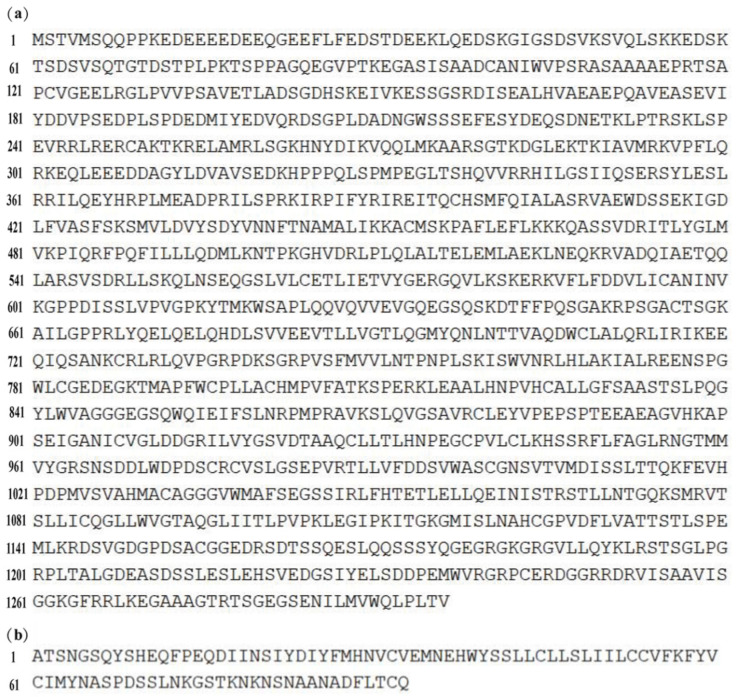
Comparison of amino acid sequences of RhoGEF10 in males and females (Appendix A). Note: (**a**): Amino acids of RhoGEF10 in female individuals; (**b**): Amino acids of RhoGEF10 in male individuals.

**Figure 6 genes-13-01262-f006:**
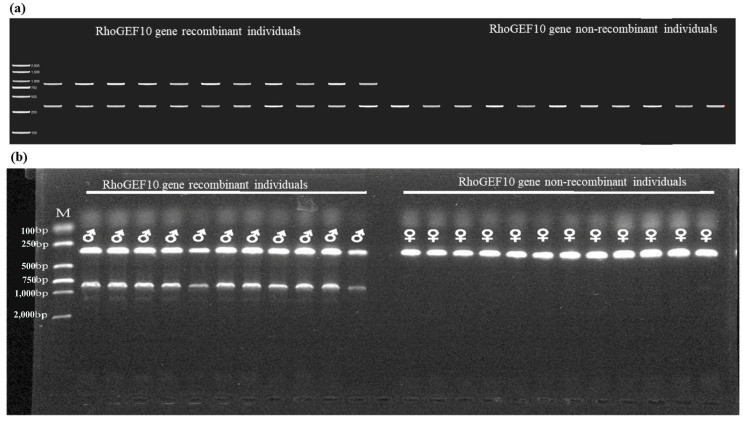
The results of PCR products of male and female *O. punctatus.* Note: M: DL 2,000 DNA Maker; ♂: physiological male fish; ♀: physiological female fish. (**a**): Electrophoresis pattern drawn with Tbtools; (**b**): 1.5% agarose gel electropherogram.

## Data Availability

The data used in the present study are available from reasonable request from the corresponding author.

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
