# Peer review of "Identification of Male-Specific Molecular Markers by Recombination of RhoGEF10 Gene in Spotted Knifejaw (Oplegnathus punctatus)"

_genes, 2022, doi:10.3390/genes13071262_

Round 1

Reviewer 1 Report

The authors in the paper entitled Identification of Male-Specific Molecular Markers by Recombination of RhoGEF10 Gene in spotted knifejaw (Oplegnathus punctatusidentified a PCR based sex determination method in spotted knifejaw fish to differentiate males from females. Such molecular markers are important for identification of sex in aquatic species. 

The authors claim that this method improves detection efficiency preparation of male seedlings, the PCR based technique that is discussed in the paper is based on muscle sampling. For seedling purposes, the authors need to justify how this technique will be used to identify only males. Even though the marker was identified it seems that the manuscript lacks clarity in the application of identified marker for claimed applications. 

Lines 61-75: This information do not have any relevance to the current study. The introduction should be more focused to the current study.

Lines 86-93: provide abbreviation and links for the tools used.

Line 105: it was mentioned the OD260 nm OD230 ≥1.8, OD260 nm OD230 nm≥1, Indicate that these values are the ratios “OD260/ OD230≥1.8”. The OD260 nm/OD230nm ratio should also be more than 1.8 for a good quality DNA not just more than 1

Lines 121-123: State the number of PCR cycles clearly. 

Figures – Include the note section of Figures to the Figure Legend.

Show the mapping of RHOGEF10 region localization of sex and somatic chromosomes using chromosome collinearity map. It is necessary to show that this gene is exclusively mapped to sex chromosomes.

Why did the protein in the mutated RhoGEF10 shorten after insertion? Was the insertion in the splice junction?

The discussion and conclusions claim that the greater significance of this gene in peripheral nerve conduction patterns and myelin 283 development between female and male. Experimental evidence is not provided for such claims. Hence these claims need to be discussed carefully. Besides, the mutated gene on Y chromosome is compensated by the gene on X-chromosome, so the claims made were ambiguous with no clear explanation of experimental evidence. 

Author Response

Dear Reviewer:

  Thank you very much for your time to review our manuscript and pointing out the problems and giving advice for improvements, especially in terms of article writing, grammatical and formatting corrections. These suggestions will be of great help to our future research as well. We have made the corresponding corrections and explanations in response to the queries one by one in the annex. At the same time, we have reviewed the entire text for grammatical and spelling errors.

Reviewer 2 Report

I have gone through the article. Each part is well written. The figures are well represented. The paper is recommended to publish.

Author Response

Dear Reviewer:

  Thank you very much for your time to review our manuscript. We express our sincere gratitude for your recognition and wish you a happy life!